# Trace Element Bioaccumulation in Stone Curlew (*Burhinus oedicnemus*, Linnaeus, 1758): A Case Study from Sicily (Italy)

**DOI:** 10.3390/ijms21134597

**Published:** 2020-06-28

**Authors:** Chiara Copat, Margherita Ferrante, Béatrice V. Hernout, Flavia Giunta, Alfina Grasso, Andrea Messina, Rosario Grasso, Maria Teresa Spena

**Affiliations:** 1Department of Medical, Surgical and Advanced Technologies “G.F. Ingrassia”, University of Catania, Via Santa Sofia 87, 95123 Catania, Italy; ccopat@unict.it (C.C.); marfer@unict.it (M.F.); agrasso@unict.it (A.G.); 2Institute for a Sustainable Environment, Clarkson University, 8 Clarkson Ave, Potsdam, NY 13699, USA; bhernout@clarkson.edu; 3Department of Biology, Clarkson University, 8 Clarkson Ave, Potsdam, NY 13699, USA; 4Department of Biological, Geological and Environmental Science, University of Catania, Via Androne 81, 95124 Catania, Italy; messinaandrea.90@gmail.com (A.M.); rosagra@unict.it (R.G.); marisaspena@hotmail.com (M.T.S.)

**Keywords:** metals, environmental monitoring, birds, ecotoxicology, blood, feathers

## Abstract

The study aimed to highlight the degree of trace element contamination along three sites of Sicily: the Magnisi peninsula (MP), located in proximity to the Augusta–Priolo–Melilli petrochemical plant; the Ragusa agro-ecosystem (RA), characterized by a rural landscape; and the Gela plain (GP), characterized by intensive agriculture and a disused petrochemical plant. We collected biological samples (abraded back feathers and blood) of the Stone Curlew (*Burhinus oedicnemus* Linnaeus, 1758) as well as soil samples to determine the trace elements concentrations of As, Cd, Co, Cr, Cu, Hg, Mn, Ni, Pb, Zn, Se and V using ICP-MS analysis. The results found for the three sites show different trends of accumulation, which depend on the different management and geological characteristics of the areas. The Gela plain and Magnisi peninsula showed a higher degree of contamination (As, Co, Cu, Mn and Se for the Gela plain; Pb and Hg for the Magnisi peninsula). Nevertheless, no critical values were found for either the environment—if the results are compared with the legal limits fixed by the Legislative Decree No. 152/2006, approving the Code on the Environment—or for living organisms—if the results are compared with the toxicological thresholds for birds, especially if the short-term exposure results from the blood values are considered. Only the Se levels in animal blood from the RA and GP were found slightly higher than the minimum level required in bird diets. The positive scenario can be attributed on the one hand to the interruptions of emissions of the Gela refinery around 5 years ago, and on the other hand to the more intense and strict controls that are implemented in the area surrounding the petrochemical pole of Augusta–Priolo–Melilli.

## 1. Introduction

Environmental pollution is a topic of high interest, especially due to the drastic increase in pollutants released into the environment observed through the last decade. These pollutants may have the potential to accumulate in soil, water and air [1,2,3]. Various vertebrate organisms, such as fish, bats and birds [4,5,6], have been useful biomarkers of environmental contamination. Hence, a large number of studies on plants and animals from polluted areas have been carried out [7,8]. 

Non-essential metals, such as cadmium (Cd), lead (Pb) and mercury (Hg), as well as metalloids, such as arsenic (As), represent important types of pollutants since they are particularly persistent in the environment [7,8] and can be harmful to organisms when taken up in small amounts [9,10,11]. Essential metals, such as copper (Cu), zinc (Zn) and manganese (Mn), are considered essentials for maintaining a good metabolism. Whereas the deficiency of essential metals can also elicit negative effects, an excess of these metal concentrations in the organism may elicit several toxic effects [12,13]. 

Metals can be adsorbed by the organisms through various exposure routes, among which food exposure is usually the most important, followed by dermal exposure and inhalation [14,15,16]. Some organisms are able to bioaccumulate an amount of metals in their tissues or organs without observing any adverse effects; however, once the internal metal concentration reaches a toxic threshold, adverse effects may occur [4,17]. 

Many studies about metal bioaccumulation have focused on birds [18,19,20,21,22,23,24,25,26,27,28,29,30,31,32], showing their relevance as biological indicators of pollution. Birds are suitable bioindicators of metal pollution due to (1) their diverse ecology, which allows to show the effects of trace elements on multiple levels of the ecosystem; (2) their ability to bioaccumulate metals in their feathers, blood and other tissues; (3) their ubiquity and worldwide repartition; and (4) their sensibility to a wide range of pollutants, by showing adverse effects after various level of exposure: for instance, impairment of reproduction and growth, endocrine and nervous disorders, genotoxicity, and physiological and behavioral abnormalities affecting survival [33]. 

In previous studies illustrating the effects of metals in birds, the effects of certain pollutants on birds across different habitat and ecological systems were analyzed [18,19,20,21,23,24,25,26,27,28,29,30,31,32,34]. For non-essential toxic trace elements, toxic thresholds have been investigated. These thresholds are the levels of trace elements above which adverse effects occur, such as damage to development, reproduction and the immune system in the case of arsenic; asymmetry of wing feathers in the case of mercury [19]; or sublethal effects, such as problems with thermoregulation and movement due to lead exposure [35]. However, there is a lack of data for certain bird taxa, such as the Burhinidae family. 

This is the first paper presenting the potential bioaccumulation of metals in the Stone Curlew (*Burhinus oedicnemus* Linnaeus, 1758), as well as in the Burhinidae family. This species was selected for this study as a bioindicator of metal pollution since this Charadriiformes lives in wide, flat and dry areas, such as open farmed crops or pastures, but it can be found also near wetlands, especially on pebbly riverbeds. It often lives in proximity of anthropic contexts, and it nests on the ground. The Stone Curlew is included in the IUCN Red List as Vulnerable C1, and is listed in Attachment I of the Birds Directive (2009/147/CE). Its critical conservation conditions are related to the increasing urbanization, as well as the agricultural mechanization and the use of pesticides (IUCN, 2019). Therefore, this species could be exposed to various anthropogenic pollutants, including metals. In this study, we proposed to use an animal model little studied, as an indicator of metal pollution in some areas of Sicily that have showed issues related to the management of the environment. Therefore, we aim to convey the attention of the scientific public—but not only—to those emergencies that would most likely adversely affect the ecosystem in the long term.

By using the selected bioindicators species, this work aims to identify a pattern of trace elements contamination between areas of Sicily with a different anthropic resource exploitation, highlighting differences in the short- and long-term exposure and assessing if the trace element concentration bioaccumulated in the biological tissue of Stone Curlew may be harmful to their health status. The general purpose of the study is to analyze the trace elements present in different tissues, and to assess the relationships between the concentrations in the animal’s tissues versus the environment (soil matrix).

To reach our aim, we (1) assessed the bioaccumulation of twelve elements, namely arsenic (As), cadmium (Cd), cobalt (Co), chromium (Cr), copper (Cu), mercury (Hg), manganese (Mn), nickel (Ni), lead (Pb), zinc (Zn), selenium (Se) and vanadium (V), in feathers (*n* = 93) and blood (*n* = 87) of the Stone Curlew; (2) compared the amount of metals accumulated in the bird tissues and ground samples between three sites; and (3) compared the levels of metals observed with the toxic thresholds previously established for non-essential metals and metalloids, as well as the upper-range values for essential metals in birds. 

## 2. Results

The results show a preferential accumulation of metals in the abrased back feathers, while the concentrations found in blood are generally lower, with the exception of selenium (Figure 1 and Figure 2). See Appendix A for the results obtained for each type of sample in the three areas.

For the abrased back feathers, we found several trends of metal distribution. Overall, our data indicate higher metal concentrations in feathers sampled in the Gela plain for As, Co, Cu, Mn, Se and V when compared to the other sites, and higher metal concentrations for Pb, Cd and Hg in the Magnisi peninsula and Ragusa agro-ecosystem. In particular, for some elements such As, Mn, Se and V, the concentrations were significantly higher in the Gela plain samples than in the Magnisi peninsula and Ragusa agro-ecosystem (*p* < 0.001). Co was significantly higher in feathers collected in the Gela plain (*p* < 0.01 vs. Magnisi peninsula; *p* < 0.05 vs. Ragusa); Cu in the Gela plain showed a similar trend compared to the Ragusa site (*p* < 0.05). For Pb, the bird feathers sampled in the Magnisi peninsula had higher concentrations than in the birds sampled in the other sampling sites (*p* < 0.001 vs. Ragusa agro-ecosystem and Gela plain). Cd and Hg showed a higher concentration in feathers gathered in the Magnisi peninsula and in the Ragusa agro-ecosystem compared to the Gela plain samples (*p* < 0.01). No significant differences were found in Cr, Ni and Zn accumulation in feathers between the three areas.

Blood accumulation followed a comparable trend across the three sites. As, Cu, Mn and V showed a higher concentration in the Gela plain samples compared to the Magnisi peninsula and Ragusa agro-ecosystem ones (*p* < 0.05 or *p* < 0.001 vs. Magnisi peninsula and/or the Ragusa site). Co was found higher in the Ragusa samples (*p* < 0.01 vs. Magnisi peninsula; *p* < 0.05 vs. Gela plain), while Se showed a preferential accumulation in the Ragusa samples compared to the Magnisi peninsula ones (*p* < 0.01), but in a lesser amount compared to the Gela plain samples (*p* < 0.001 vs. Magnisi Peninsula; *p* < 0.05 vs. Ragusa). No significant differences were found in Cd, Cr, Hg, Ni, Pb and Zn accumulation in blood between the sites. 

Positive correlations were observed between the element concentrations in feathers and blood (among all the individuals) for V, Mn, Hg, As and Se (*p* < 0.05) (Figure 3).

Table 1 and Table 2 show the comparisons between the levels of pollutants detected in tissues and the thresholds found in the literature. Note that the blood concentrations are expressed both in mg/kg w.w., in mg/kg d.w. and in ng/mL, since in the literature this tissue is not expressed using a unique dimension.

No studies reporting thresholds related to Co and V in feathers, as well as Co, Cr, Ni, V and Zn in blood were found. 

Regarding trace elements in the soil (Figure 4), the As concentrations were found to be higher in soils collected in the Magnisi peninsula and Gela plain (*p* < 0.05 vs. the Ragusa agro-ecosystem). Cd, Cr, Ni and V concentrations were found higher in soils collected in the Magnisi peninsula and Ragusa agro-ecosystem (*p* < 0.01 vs. the Gela plain); Mn and Co concentrations were higher in the Gela plain and Ragusa agro-ecosystem vs. Magnisi peninsula (*p* < 0.05 and *p* < 0.01 respectively); Cu concentrations were higher in the Gela plain (*p* < 0.05 vs. the other sites). Hg, Pb and Zn showed a significant accumulation in the Magnisi peninsula samples (*p* < 0.01 for Hg and Pb and *p* < 0.05 for Zn vs. Ragusa and the Gela plain). 

No significant differences in the accumulation of Se in soils were found between the three areas.

## 3. Discussion

Among the three sampling sites, the Gela plain showed the highest bioavailability of As, Co, Cu, Mn and Se in both the biological and abiotic samples. In the soil, the 75° percentile of the As concentration (22.8 mg/kg d.w.) and the 75° percentile of the Se concentration (3.7 mg/kg d.w.) are slightly above the legal limits (respectively 20 and 3 mg/kg d.w.), as fixed by the Legislative Decree No. 152/2006, approving the Code on the Environment (see Appendix A). The Se concentrations in the blood of animals from the three sites do not differ significantly from each other; nevertheless, the values found in the Gela plain and Ragusa agro-ecosystem could be of concern since they are higher than the minimum level required in the goshawk (*Accipiter gentilis*) diet according to Stout et al. (2010) [45], ranging from 0.130 to 0.200 mg/kg w.w. It can therefore be deduced that the environmental availability of Se in the three stations studied is higher than the amount necessary for the animal’s dietary needs. Se is an essential element, but an excessive intake could induce Se poisoning and results in various health problems, such as promoting inflammation and impairing the immune function, as was observed in an in-vivo study in chicken spleens [46]. In humans there are also several associations with high selenium in serum and hair, with a number of adverse health endpoints, such as a higher prevalence of nausea and vomiting, bad breath, worm infestation, breathlessness during exertion, bad breath, chest pain, hair and nail abnormalities and loss, garlic odour, oedema, spontaneous abortion and overall selenosis [47]. Nevertheless, its level of toxicity may depend on its chemical form, as inorganic and organic species have distinct biological properties [48]. High exposure to inorganic selenium in drinking water was found to be associated with a high incidence of amyotrophic lateral sclerosis (ALS) in a Northern Italian community [49].

Furthermore, in biological samples from the Gela plain, there were also higher concentrations of V and Ni (the last not significantly higher) compared to the other two sites, although the Magnisi peninsula and Ragusa agro-ecosystem showed the highest concentrations in the soil.

Among the elements found in the Gela plain, Co, Cu, Mn, Ni and, as mentioned earlier, Se, are the most essential elements because they are involved in important enzymatic reactions, but if they are present in high concentrations they are considered toxic [50,51,52]. These metals are present in numerous pesticide formulations, such as pesticides, fungicides on vines compost, and fertilizers or sprays against ornamental plant pests [53,54]. The most toxic metalloid found in the Gela plain is As. Animal and human exposure to As is a public health issue because of its systemic effects, such as gastrointestinal, neurological, reproductive, liver, kidney and blood-related disorders, as well as carcinogenicity from chronic low-dose exposures [55]. The bioavailability of As seems to refer not only to its past use in fungicides, herbicides, insecticides, etc., as well as in phosphate and organic fertilizers [56], but also to the previous pollution of the groundwater, since once oxidized, As is very soluble in water [57]. Nevertheless, the values of As found in the blood samples are three-fold below the toxicological limits defined by Burger and Gochfeld (1997) [43], and the values found in feathers are seven-fold below the toxicological limits defined by Eisler (1988) [36]. However, it is important to emphasize that for some elements, such as As and Se, the identification of the chemical form(s) (species) is of crucial importance as this form critically influences their availability and biological effects in living organisms [55,58].

The contamination degree revealed on the Gela plain derived from the marked anthropization due to the presence of inhabited centres and a petrochemical plant disused a few years ago, which since August 2019 was converted from a conventional fossil-fuel refinery into a bio-refinery to produce high-quality, cleaner fuels. The presence of V and Ni seems to be related to the specific industrial processes of the past, because they represent the main contaminants originating from oil activities or oil spills, recognized to cause detrimental effects to living organisms [59]. Ni and V are available in the majority of crude oil sources (generally at much higher levels than any other metal); hence, they can be indicators of crude oil pollution [60]. Nevertheless, the area is also subjected to intense agricultural exploitation, with several greenhouses; thus, it suffered a repeated use over time of substances such as herbicides and pesticides. All these factors have caused a dispersion of pollutants into the ecosystem, such as arsenic, mercury, nickel, lead, cadmium, chromium, antimony, vanadium, hydrocarbons, benzene, toluene and other toxic and carcinogenic organic compounds [61]. Consequently, this area of southern Sicily is known for its high mortality rate linked to tumour diseases and congenital malformations, and since 1990 the area of Gela municipality was included among areas at high risk of environmental crises. In 1998, an extensive area of Gela municipality was declared a Site of National Interest (SNI) for soil remediation [61]. 

Unexpectedly, several similarities were found between the sites of Magnisi peninsula and Ragusa agro-ecosystem regarding the bioaccumulation of the toxic metals Cd and Hg in feathers, as well as the higher concentrations of Cr, Cd and Ni in the soil, although all the values were well below the thresholds of concern for the environment and organisms’ health. Cadmium is a metal mainly used in agriculture, sewage sludge and phosphate or organic fertilizers [62]; it may be highly toxic at low concentrations, and tends to accumulate preferentially in the soft tissues, from which it is insufficiently eliminated—although, in nature the toxic phenomena linked to Cd are more prevalent between aquatic species than among terrestrial ones [63]. Mercury often derives from industrial production processes and waste combustion, but in this particular case it should be noted that Hg discharging into the sea from the Rasiom refining plant in Augusta has occurred in the past [64]. The main concern regarding the accumulation of Hg derive from the following characteristics: the less toxic forms of the element can be transformed into more toxic forms through biological processes (as methylmercury) [65]; Hg can be bioaccumulated and biomagnified through the food chain [65]; and it is a mutagenic, teratogenic and carcinogenic metal and causes histopathological mutations [66].

Despite the high toxic potential of Cd, Cd concentrations in feathers proved to be very low in all the sites (0.032 mg/kg d.w. for the Magnisi peninsula, 0.029 mg/kg d.w. for the Ragusa agro-ecosystems and 0.010 mg/kg d.w. for the Gela plain) compared to the thresholds defined by Burger (1994) [37], which suggests adverse effects in some bird species above 0.1–2 mg/kg d.w.; by Stock et al. (1989) [38] and Meador (1996) [41], which suggests kidney damage above 2 mg/kg d.w.; and by Spahn and Sherry (1999) [67], who noted that the same cadmium concentration lead to a reduction of bone growth rates. Concerning the use of feathers for Cd biomonitoring in birds, there is some uncertainty in the scientific community. In fact, while some studies show that the preferential source of Cd contamination in birds is external exposure [68], others report that Cd is preferentially seized from internal sources, with few high concentration external particles [69]. For Hg, a limit for the asymmetry of the flight feathers at 40 mg/kg w.w. was defined [19], and the occurrence of sublethal effects at 5 mg/kg w.w. [39] and at 5 mg/kg d.w. [35]; our values have fallen far below the aforementioned limits (0.416 mg/kg d.w. for the Magnisi peninsula, 0.376 mg/kg d.w. for the Ragusa agro-ecosystems and 0.193 mg/kg d.w. for the Gela plain).

The choice for the type of feather to be analysed was based on previous bibliographical data, which showed that older feathers absorb a greater quantity of trace elements due to their longer lasting contact with the animal’s tissues and the environment. In addition, other studies showed that the stage of the moult can influence the absorption of metals from feathers [70,71]. During the moult, the levels of some metals in the internal tissues of the birds decrease as they are sequestered in the feathers. After completion of the moult cycle, the levels of some metals rise in the internal tissues until the next moult [72]. Consequently, feathers that are growing first should contain the highest metal concentrations, while feathers that are growing last should have accumulated the lowest concentrations [73].

In the Stone Curlew, moulting begins around April to mid-June and ends in September–November [74,75], although it should be emphasized that these intervals provide rather generic indications, at least in part due to the lack of information so far available. In any case, it should be noted that the uncertainty of the estimates reported in the literature could also reflect a significant variability in the timing of moulting in different populations and, above all, at different latitudes. The populations investigated in this study occupy reproductive sites as early as the middle of February.

As for the back feathers moult, it occurs gradually, and for the purposes of the study we have focused on the abraded feathers to be sure that they have stayed in contact with the animal for longer. Future studies involving the analysis and comparison of other types of feathers in the Stone Curlew could provide further details.

While feathers can provide information regarding the relative long-term accumulation, blood samples—as a frequently renewed and filtered matrix—provide information about the short-term absorption of elements by the individual. Our results revealed the same short-term exposure of Cd and Hg in all the considered areas of Sicily. For Cd in blood, a low threshold of 0.5 ng/mL is associated with alterations in the activity of antioxidant enzymes in the griffin (*Gyps fulvus*) [22]; results we found are 1.25-fold lower than this threshold but also below our calculated limit of quantification (LOQ) [76]. As for Cd, concentrations of Hg in the blood did not differ significantly between the sampling area and, in the Ragusa agro-ecosystem, Magnisi peninsula and Gela plain, it was found with concentrations, respectively, of 1.03-, 1.20- and 1.33-fold lower than the toxic threshold of 30 ng/mL associated with alterations in antioxidant enzymes activity of *G. fulvus* [22], but significantly lower (about 30 times less) than the toxic threshold of 1–3 mg/kg w.w. associated with reproductive failure in birds [19].

Certainly, the sources of contamination are different. The landscape situation of the Ragusa agro-ecosystem is characterized by typical rural and pasturable lands, similar to other south-Mediterranean areas, such as Creta or Tunisia, which also show fragmented cultural mosaics of cereal crops, woodlands and pastures [77]. The area of the Magnisi peninsula presents several environmental issues, among which the most striking are the proximity to the petrochemical pole of Augusta–Priolo–Melilli and the insistence of a dump of pyrite ash. Here, industrial activities started in the late 1950s and caused progressive contamination of the environment over the years. Previous studies have reported high contamination in the marine environment [17,78,79,80,81,82], showing severe metal (Cd, Cr, Cu and Hg), PCBs and PAHs contamination. Nevertheless, terrestrial organisms, such as the ground beetle *(Parallelomorphus laevigatus*) or the sandhopper (*Talitrus saltator*) living in the Magnisi peninsula, or on the nearby beach of Marina di Priolo, did not show critical concentrations of trace elements when compared with other Sicilian areas [83,84], with the exception of Pb and Hg bioaccumulation in *Ligia italica* [85].

According to our results, the ones that distinguished the Magnisi peninsula from the other sites are the higher concentrations of Pb in feathers. Lead is a metal with various industrial uses: it is found in batteries, in buried electrical cable coverings, in pipes for gas and water pipes (where it is used as a stabilizer for PVC), in welds, leaded pottery, and its past use in the printing industries, manufactories, plumbing and gasoline industries is well known [86]; it is plausible that the abundance of this metal in samples from the peninsula is correlated with human activities in the area, including the presence of numerous roads, considering the past extensive use of the element in fuels. It is known that Pb could interfere with different metabolic systems, particularly in the forms of chronic intoxication and as a mutagenic, carcinogenic and teratogenic agent, while it has no physiological role [87]. Moreover, the absorption of Pb is influenced by the age of the animal [41]. Despite this, in this study, the Pb concentrations determined in the blood are 5-fold below the toxicological threshold proposed for feathers in birds [35], and the short-term exposure indicated by Pb in the blood did not highlight differences between the sites, which are characterized by values significantly lower than the proposed toxic thresholds [22,44]. In the Magnisi peninsula, higher concentrations of Hg, Pb and Zn in the soil were also found; nevertheless, the concentrations are below the limit fixed by the Legislative Decree No. 152/2006, although the 75° percentile of the Se concentrations (3.6 mg/kg d.w.) from the Magnisi peninsula, as we found in the Gela plain, is slightly above the legal limits.

We can also note that the Cr values, in the abrased back feathers of all the stations, are a little above the 2.8 mg/kg d.w. limit reported by Burger and Gochfeld (2000) [35]. In particular, the average concentrations are higher in the Magnisi peninsula site and in the Ragusa area, with 2.86 mg/kg d.w. and 2.74 mg/kg d.w., respectively, rather than the Gela plain, where there is an average of 2.45 mg/kg d.w. It is important to underline however that the analysed Cr is referred to the total and not to a particular species. It is well known that the toxic species is the hexavalent one (CrVI), and it would therefore be interesting to perform a speciation of the metal to accurately quantify the concentrations of the toxic metal. Chromium, as Cd, is mainly used in agriculture; hexavalent chromium at high concentrations is associated with carcinogenic activity and important toxic effects, while the trivalent one—on which we focused—is not particularly toxic to organisms [88]. 

As for soil, the samples collected were too few in order to make a certain comparison with the biological tissue samples (*n* = 15, 5 for each sampling area), but the results show a corresponding trend in As and Mn concentrations between the biological tissues and soil samples. In fact, these metals showed a statistically significant preferential accumulation both in the soil samples from the Gela plain and in the blood and feathers samples taken by individuals from the same study area. For the reasons described above, it is possible that As has a discrete presence in the soil of the Gela plain, and this may be reflected in the feeding of the Stone Curlew; however, note that the preferential accumulation of As in soils is statistically significant, also in samples taken from the Magnisi peninsula, where the same is not true in the biological samples. In the same way, Mn appears to have a preferential accumulation in soils taken from both Gela and Ragusa. This may have to do with the management of the two territories, mainly used for intensive crops that require the use of pesticides, but correlations can only be noted with the biological samples of the birds of the Gela plain, and not with those of the Ragusa agro-ecosystem.

Furthermore, we cannot exclude the possibility of external contamination of the feathers by metals, a hypothesis reinforced by previous studies [70]. This contamination can occur both because of direct atmospheric deposition, and because of preening conducted by the animal itself through its beak [89].

Further studies carried out by focusing more on soil samples could clarify the relationship between the metal content of the soil and the type of metal absorbed by the Stone Curlew.

## 4. Materials and Methods

### 4.1. Sampling Areas

Among the different habitats of Sicily used as nesting sites by the Stone Curlew, we selected three areas with different management strategies, levels of anthropic influences, and environmental issues (Figure 5).
The Magnisi peninsula (MP), located in the south-eastern coast of Sicily between the Augusta and Siracusa gulfs, a few kilometers away from the residential area of Priolo Gargallo (SR), has a rectangular shape with long sides, parallel to the coast, and about 2 km long with the short sides about 1 km. For more informations about geomorphological aspects refer to literature [90]. The peninsula has a flat surface that reaches the maximum height of 19 m above sea level. It shows a bare ground with sporadic buildings and wild fields, involved in cow pasture since the beginning of the last century. The proximity to the Augusta–Priolo–Melilli petrochemical plant, as well as the presence of a pyrite ashes dump, pose a threat to the environmental health of the area. Despite these issues, the site shows naturalistic peculiarities, for instance the presence of endemic plants, such as *Limonium syracusanum* Brullo, as well as a bird biodiversity ranging from migratory to non-migratory species [91]. In particular, the population density recorded for the Stone Curlew is far greater compared to the other southern regions of Italy [91,92]: it depends on the presence of bovine excrements, which turn into a substantial source of invertebrates on whom the birds feed, and therefore cause a decrease in intraspecific competition [93].The Ragusa agro-ecosystem (RA) is characterized by a rural landscape, where arable lands delimited by dry stone walls alternate with the association of olive and carob trees (*Oleo sylvestris–Ceratonion siliquae* Br.- Bl. ex Guinochet and Drouineau). The woodlands overlooking the canyons represent an excellent refuge for several vertebrate taxa, as well as an important nesting site for numerous species of birds.The Gela plain (GP), with an extension of 447.8 km^2^, is the second largest alluvial plain in Sicily and southern Italy in terms of size. It features an extensive agricultural mosaic, in which the Stone Curlew nests in the cultivated crops areas (56%, mainly cereals), artichokes land (22%), abandoned pastures and garrigues (17%) and olive tree groves (3%). The anthropic pressure on the area is focused in particular on the coast while the fields, subject to intensive farming and full of irrigation canals, are sprayed with pesticides, fertilizers and herbicides. We chose the Gela Plain due to the presence of a petrochemical plant, abandoned since 2014, with the aim of identifying possible relationships between pollution and the use of the area. Moreover, the rural area and its surroundings are crossed by several provincial and rural roads. On the other hand, there are also semi-natural environments, such as olive groves, wetlands, pastures, garrigues and small woods. The alternation of these habitats increases the fauna and flora communities.

All the sites described above host nesting populations of Stone Curlew, whose presence over the years has been recorded [77,92,93]. We have chosen these three areas for their different types of environmental situations and land management, which will consequently lead to a different type of pollution. Due to their proximity to polluted areas and industrial anthropic activities, we expect the Magnisi peninsula and the Gela plain to be more polluted than the Ragusa agro-ecosystem.

### 4.2. Sampling

Sampling was performed under the authorization from Ispra (Higher Institute for Environmental Protection and Research), a public research body supervised by the Ministry of the Environment and Protection of Land and Sea (Italy), to collect samples of bird’s feathers and blood for scientific purposes (No. 44648, Date 11/07/2018).

Samples (feathers and blood) were taken during the Stone Curlew’s breeding season. In the study area of the Magnisi peninsula and the Ragusa plateau, sampling was carried out in 2018 and 2019, while in the Gela plain only in 2019. We chose to analyse the accumulation of metals in abrased back feathers because of their long persistence on the bird’s body, and their consequent capability to stock metals for a longer time compared to other feather typologies [94]. On the other hand, the analysis of the metal content in blood provides information about the accumulation of pollutants within a short time after the uptake [95]. 

The collection of samples is closely linked to the identification of the nesting site, since in the chosen areas it is not possible to capture the specimens with the mist nets. The nests containing eggs were individually identified, then a specific trap was set on each nest to capture the bird while brooding. It is important to specify that all the activities were carried out causing the least possible harm to the animals; they were handled with caution and released immediately at the end of the samplings. From each bird, samples of abrased back feathers (about 250 mg w.w.) (*n* = 93), as well as 2 mL of blood (*n* = 87) were collected. Moreover, a ground sample of the soil matrix (about 250 mg w.w.) (*n* = 15) near each nest was collected. Blood was collected in heparinated tubes to prevent coagulation and stored in a portable refrigerator to avoid deterioration. Next, the bird was released near the nest. Samples were stored in a freezer at -80 °C until further analysis. We captured a total of 96 individuals, all of which were adults, and collected 195 total samples, including the feathers, blood and soil matrix.

### 4.3. Trace Element Extraction and Analysis

The feather samples were rinsed three times with double-distilled water in Falcon tubes following the method described in Markowski et al., 2013 [21].

All the samples were weighted by the use of an analytical balance (Mettler Toledo) (feathers and ground samples ≈ 0.25 g; blood ≈ 1 mL). Next, samples were dried overnight at 60 °C, and weighted a second time to determine the dry weight. The resulting water contents were 12.2%, 14.7% and 78%, respectively, for the feathers, ground samples and blood, and were used to convert the results from d.w. to w.w. to aid a literature comparison. Samples were mineralized in a microwave oven (Ethos, TC, Milestone) with a mixture of strong acids. To digest the feathers, a solution was prepared with 6 mL of 67% superpure nitric acid (HNO_3_—Carlo Erba, Milano, Italy) and 2 mL of 30% peroxide hydrogen (H_2_O_2_—Carlo Erba, Milano, Italy) and samples were digested for 35 min at 120 °C. Blood samples were digested using a solution of 3 mL of 67% superpure nitric acid and 800 µL of 30% peroxide hydrogen, for 30 min at 80 °C. Soil samples were digested with a solution of 3 mL of 67% superpure chloridric acid (HCl—Carlo Erba, Milano, Italy) and 9 mL of 67% superpure nitric acid, for 35 min at 130 °C.

After acid digestion, for the feathers and ground samples, the content of the vessels was decanted in Falcon tubes and double-distilled water (Merck, Kenilworth, NJ, USA) was added to the samples up to 50 mL, while for the blood samples, water was added up to 20 mL. Concentrations were determined using standard solutions prepared in the same acid matrix. Standards for the instrument calibration were prepared based on the mono element certified reference solution ICP Standard (Merck, Kenilworth, NJ, USA). 

For quality control, a sample for each batch of mineralization was processed in duplicates; one was spiked with a multi-element solution of 25 mg/L with 5 mg/kg, and we obtained a mean recovery of 89–112% for all the metals. 

Trace element quantification was performed with an Inductively Coupled Plasma–Mass Spectrometer (ICP-MS) Elan-DRC-e (Perkin Elmer, Waltham, MA, USA). A total of 10 mL from each digested sample and the standards for the instrument calibration were added with 50 µg L^−1^ of internal standard (Yttrium, Y, and Rhenium, Re, 1000 mg/L Merck, USA) before the reading, to minimize instrumental drift. 

The limits of detection (LOD) (Appendix A) were calculated based on the 40 CFR 136 EPA procedure (US-EPA, 2016) for digested analytical blanks, using the following Equation (1):LOD = t (df = n − 1, p = 0.99%) × SD(1)
where t is the one tail student’s t distribution, df are the degree of freedom, n the number of blank replicates, and p is the probability and SD of the standard deviations.

The limits of quantification (LOQ) (Appendix A) were calculated based the following equation:LOQ = 10 × SD(2)

All the results reported for the feathers and soils were above the LOD, with the exception of one value for Se in feathers and one value for Hg in soil. Results of the trace elements in blood revealed several concentrations below the LOD, according to the following percentage with respect to the total number of blood samples analysed: 9% for As; 24% for Cd; 86% for Cr; 76% for Ni; 64% for Pb; and 26% for V).

### 4.4. Statistical Analysis

Statistical results of the metal analysis were calculated using IBM SPSS 20.0 software. The normal distribution was verified using the Kolmogorov–Smirnov test. Only the soil values did not have a normal distribution.

First, a variance analysis (ANOVA) was performed to compare the metal concentrations between sampling areas for each tissue type, with the exception of soils; then, a post-hoc Tukey test was performed to specifically determine among which areas insisted these differences. A Kruskal–Wallis test followed by a post-hoc Dunn–Bonferroni test was applied to compare the metal concentrations in the soils between sites.

A Pearson correlation test was applied to evaluate the strength of the relationships between the metal concentrations measured in feathers and blood.

## 5. Conclusions

The results found for the three sites show different trends in accumulation, which depend on the different management and geological characteristics of the areas, as well as on the peculiar characteristics of the single pollutant. Among the areas of study, the Gela plain and Magnisi peninsula showed a higher degree of contamination, although no critical values were found for either the environment or living organisms, especially if short-term exposure values are considered. The positive scenario can be attributed on the one hand to the interruptions of emissions of the Gela refinery since its closure more than 5 years ago; on the other hand, it can be attributed to the more intense and stricter controls that are enforced in the area surrounding the petrochemical pole of August–Priolo–Melilli.

Furthermore, the results obtained in this research are to be considered as pioneers of a study methodology based on the analysis of the concentration of trace metals on feathers and on the comparison with the blood values in Stone Curlew; this can represent a line of research and offer a possibility of comparison with further studies on other populations of the Mediterranean basin. 

The reasons for continuing to study this species are many and valid; the hope is that we will continue in this direction in order to have as much information as possible aimed at implementing conservation strategies to restore and manage priority landscapes, and potentially contribute to improvements in human health and wellbeing. 

## Figures and Tables

**Figure 1 ijms-21-04597-f001:**
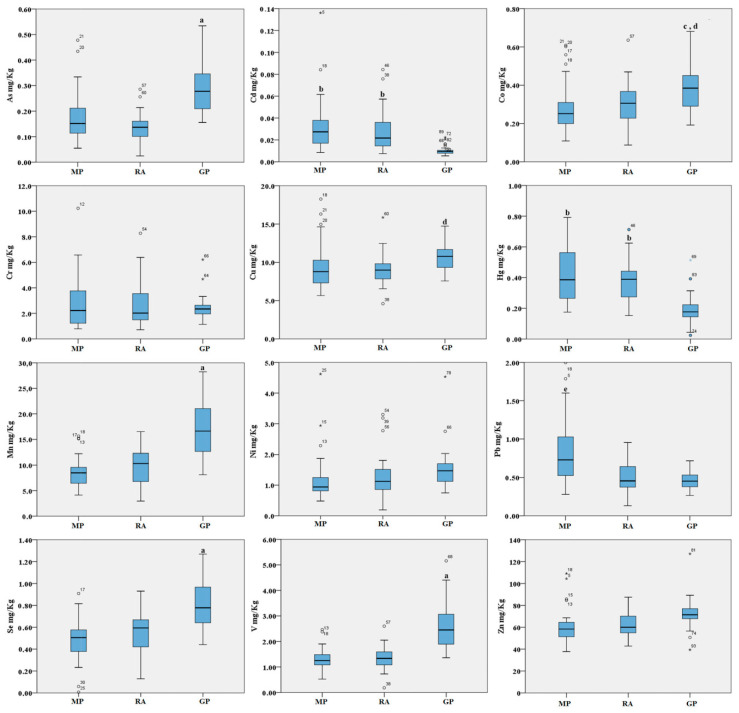
Boxplot of the trace elements in the abrased back feathers (mg/kg d.w.) of birds from the three areas. MP: Magnisi peninsula; RA: Ragusa agro-ecosystem; GP: Gela plain. The box represents the interquartile (IQ) range and contains the 1st quartile, the median and the 3rd quartile; whiskers represent extreme values of minimum and maximum; in addition, outliers with values between 1.5 and 3 times the IQ range are displayed. Legend: a = *p* < 0.001 vs. MP and RA; b = *p* < 0.01 vs. GP; c = *p* < 0.01 vs. MP; d = *p* < 0.05 vs. RA; e = *p* < 0.001 vs. RA and GP.

**Figure 2 ijms-21-04597-f002:**
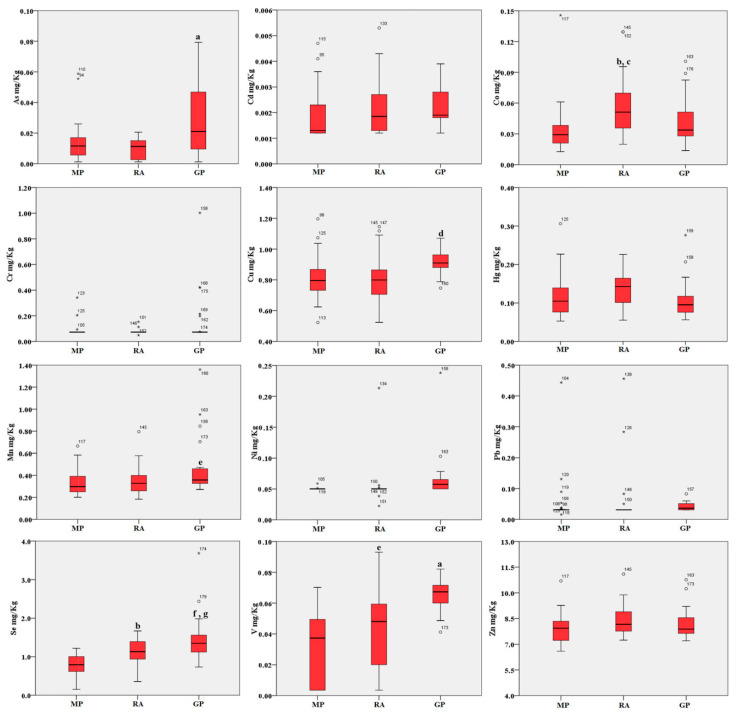
Boxplot of the trace elements in blood (mg/kg d.w.) of birds from the three areas. MP: Magnisi peninsula; RA: Ragusa agro-ecosystem; GP: Gela plain. The box represents the interquartile (IQ) range and contains the 1st quartile, the median and the 3rd quartile; whiskers represent extreme values of minimum and maximum; in addition, outliers with values between 1.5 and 3 times the IQ range are displayed. Legend: a = *p* < 0.001 vs. MP and RA; b = *p* < 0.01 vs. MP; c = *p* < 0.05 vs. GP; d = *p* < 0.01 vs. MP and RA; e = *p* < 0.05 vs. MP; f = *p* < 0.001 vs. MP; g = *p* < 0.05 vs. RA.

**Figure 3 ijms-21-04597-f003:**
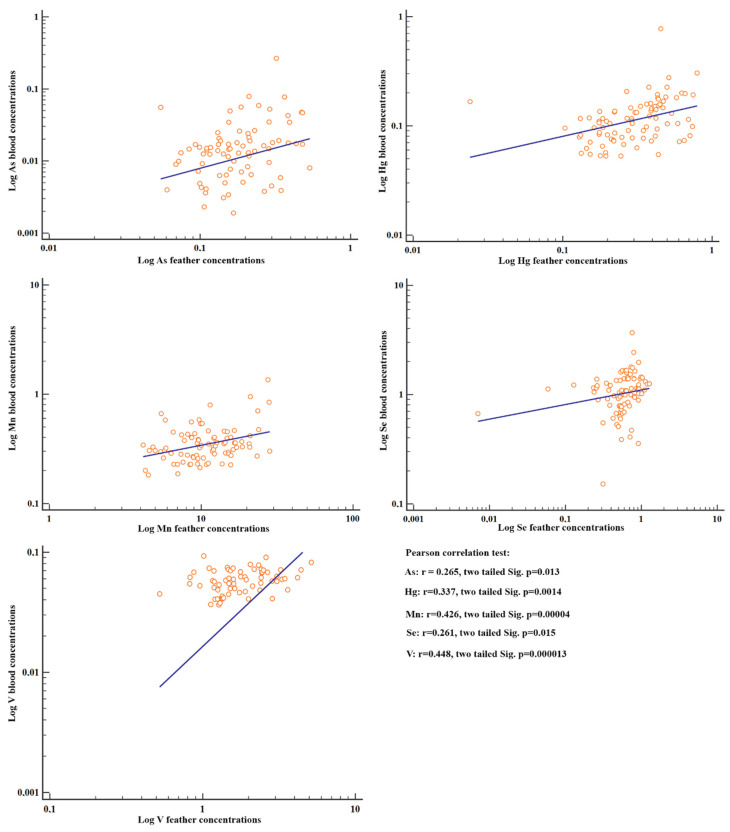
Scatter diagrams and regression lines of the metal concentrations with significant Pearson correlations between blood and feathers. Concentrations were log-transformed as log(y) ¼ a þ b log(x).

**Figure 4 ijms-21-04597-f004:**
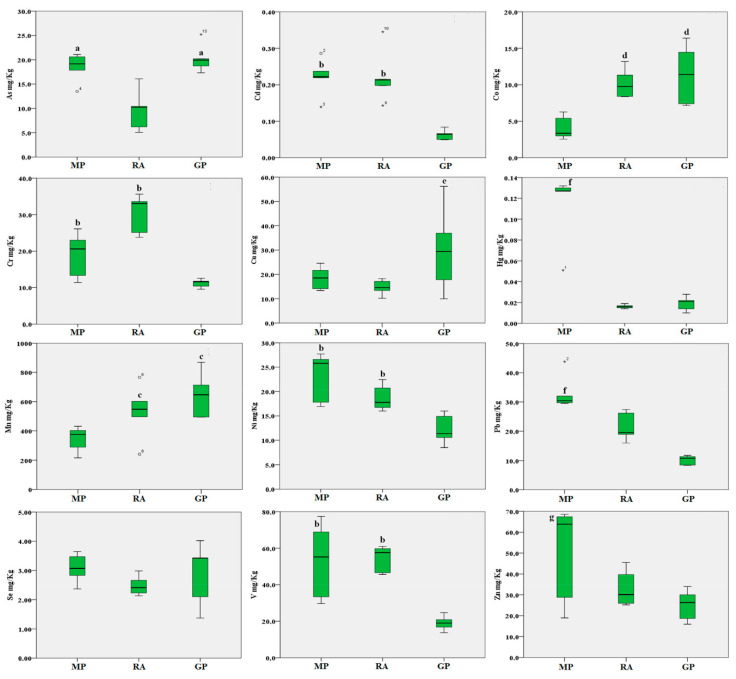
Boxplot of the trace elements in soil samples (mg/kg d.w.) from the three areas. MP: Magnisi peninsula; RA: Ragusa agro-ecosystem; GP: Gela plain. The box represents the interquartile (IQ) range contains the 1st quartile, the median and the 3rd quartile; whiskers represent extreme values of minimum and maximum; in addition, outliers with values between 1.5 and 3 times the IQ range are displayed. Legend: a = *p* < 0.05 vs. MP and GP; b = *p* < 0.01 vs. GP; c = *p* < 0.05 vs. MP; d = *p* < 0.01 vs. MP; e = *p* < 0.05 vs. MP and RA; f = *p* < 0.01 vs. RA and GP; g = *p* < 0.05 vs. RA and GP.

**Figure 5 ijms-21-04597-f005:**
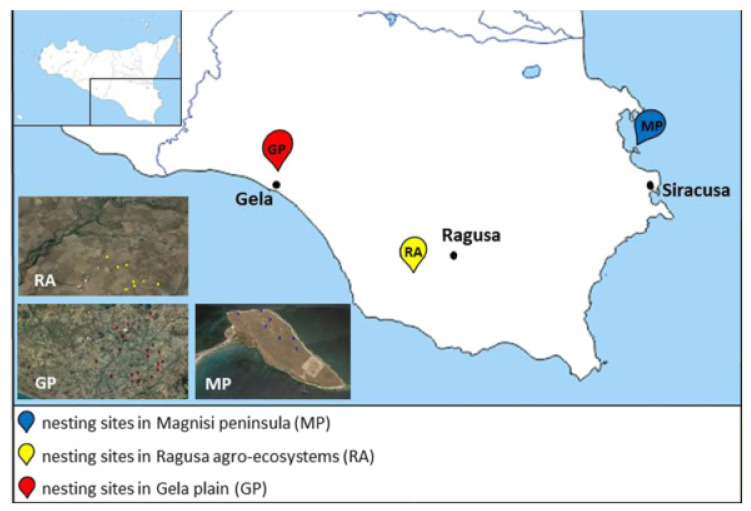
Map of the sampling sites in south-eastern Sicily. MP: Magnisi peninsula; RA: Ragusa agro-ecosystem; GP: Gela plain.

**Table 1 ijms-21-04597-t001:** Trace elements in abrased back feathers of the Stone Curlew sampled from the Magnisi peninsula (MP), Ragusa agro-ecosystem (RA) and Gela plain (GP) (in mg/kg d.w.), and the toxicological limits in other species.

Trace Elements	Magnisi Peninsula	Ragusa Agro-Ecosystem	Gela Plain	Thresholds
As	0.180	0.139	0.285	2 mg/kg w.w. [36]
Cd	0.032	0.029	0.010	0.1–2 mg/kg d.w. [37]; 2 mg/kg d.w. [35,38]
Co	0.288	0.303	0.393	
Cr	2.860	2.742	2.452	2.8 mg/kg d.w. [35]
Cu	9.420	9.007	10.76	
Hg	0.416	0.376	0.193	40 mg/kg w.w. [19]; 5 mg/kg w.w. [39]; 5 mg/kg d.w. [35]
Mn	8.564	10.06	17.24	Teratogenic effects (such as micromelia, twisted limbs, haemorrhage, and neck defects), behaviour impairments, altered growth rates and reduction of haemoglobin formation [40]
Ni	1.199	1.266	1.554	Interferences with plumage intensity [41]
Pb	0.848	0.497	0.458	4.0 mg/kg d.w. [35]
Se	0.483	0.566	0.802	5 mg/kg d.w. [42]
V	1.299	1.352	2.606	
Zn	60.85	62.09	72.56	

**Table 2 ijms-21-04597-t002:** Trace elements (mg/kg d.w.; results are also given in mg/kg w.w. and ng/mL for literature comparison) in the blood of the Stone Curlew, and the toxicological limits in other species. * The authors did not indicate if the threshold is expressed as d.w. or w.w.

Trace Elements	Magnisi Peninsula			Ragusa Agro-Ecosystem			Gela Plain			Thresholds
	B mg/kg d.w.	B mg/kg w.w.	B ng/mL	B mg/kg d.w.	B mg/kg w.w.	B ng/mL	B mg/kg d.w.	B mg/kg w.w.	B ng/mL	
As	0.014	0.003	2.982	0.01	0.002	2.13	0.028	0.006	5.964	20 ng/mL [43]
Cd	0.002	0.0004	0.426	0.002	0.0004	0.426	0.002	0.0004	0.426	0.5 ng/mL [22]
Hg	0.117	0.026	24.92	0.136	0.03	28.97	0.106	0.024	22.58	1–3 mg/kg w.w [19]; 30 ng/mL [22]
Pb	0.050	0.011	10.65	0.058	0.013	12.35	0.042	0.009	8.950	10 mg/kg * [44]; 150 ng/mL [22]
Se	0.804	0.179	171.3	1.135	0.253	241.8	1.454	0.324	309.7	1 mg/kg w.w. [42]; 0.130–0.200 mg/kg w.w. [45]

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
