# Peer review of "Trace Element Bioaccumulation in Stone Curlew (Burhinus oedicnemus, Linnaeus, 1758): A Case Study from Sicily (Italy)"

_ijms, 2020, doi:10.3390/ijms21134597_

Round 1

Reviewer 1 Report

The manuscript fits the scope of the journal and the overall research is well done. However, a minor but critical detail is missing, without which the manuscript should not be considered for publication. The manuscript lacks the permit for capturing and handling birds and the permit for animal experimentation. The first is released by Ministry of the Environment and Protection of Land and Sea; I think it is mandatory because Burhinus oedicnemus is listed in the Berna Conference as protected species. The second required authorization is provided by the Health Ministry.

A second point concerns the email address provided by the corresponding author. In my opinion, a non-private, institutional email address should be provided by the corresponding author.

Reviewer 2 Report

The MS is interesting, as it referes to monitoring of trace elements in an almost unknown species (from the toxicological point of view).

Nevertheless, the MS requires some improvement to get to a higher levels, worth of publication.

First of all, I would suggest a revision of English.

Secondly, I would also appreciate a discussion on the effect of feathers' status on trace elements concentration (it is well known that abraded, "old" feathers can have a different content of metals with respect to new feathers).

Lastly, I think that a more detailed discussion on how the values in tissues and soil correlates, and on which might be the causes.

All across the text, please state if the concentrations you report are d.w. or w.w., this will make easier for the reader to understand your results. Similarly (see file attached), please state if you made any conversion between d.w. and w.w. values, and how.

My detailed comments are in the file attached.

Author Response

Response to Reviewer 2

  1. The MS is interesting, as it refers to monitoring of trace elements in an almost unknown species (from the toxicological point of view). Nevertheless, the MS requires some improvement to get to a higher levels, worth of publication. First of all, I would suggest a revision of English.

Response: We have made some improvements to the formulation of the sentences and the lexicon.

  1. Secondly, I would also appreciate a discussion on the effect of feathers' status on trace elements concentration (it is well known that abraded, "old" feathers can have a different content of metals with respect to new feathers).

Response: I added a paragraph on this topic in the Discussion section, page 14, line 108-127.

  1. Lastly, I think that a more detailed discussion on how the values in tissues and soil correlates, and on which might be the causes.

Response: I added a paragraph on this topic in the Discussion section, page 16, line 182-200.

  1. All across the text, please state if the concentrations you report are d.w. or w.w., this will make easier for the reader to understand your results. Similarly (see file attached), please state if you made any conversion between d.w. and w.w. values, and how.

Response: Dear Reviewer, I agree with your consideration to always include how the result is expressed. Unfortunately, not all authors specify it in the materials and methods section.  However, I updated Table 1 and 2 specifying if the results are expressed as wet weight or dry weight whenever possible. As well, in materials and methods section I included the resulting water content for feathers, blood and soil, necessary for the conversion of results from d.w. to w.w.

Responses to pdf comments:

  1. Thank you for your comment. I agree that in the way it is written it is not clear the methodology used. Internal standard is used to minimize instrumental drift in inductively coupled plasma mass spectrometry (ICP-MS) and it was added to all the samples and calibration standards before the reading. I tried to improve this section in a clearer way.
  2. The sentence was modified as follows: “Results of trace elements in blood revealed several concentrations below the LOD according to the following percentage respect to the total number of blood samples analyzed: (9% for As, 24% for Cd; 86% for Cr; 76% for Ni; 64% for Pb; 26% for V)”.
  3. Toxic effects of selenium are discussed a few lines below, on page 13 from line 31 to line 41.
  4. Dear Reviewer, I found that Directive 2011/65/EU of 8 June 2011 (RoHS II Directive) prohibits the use of hazardous substances, including lead, in electrical and electronic equipment for domestic use, thus preventing the use of paints containing the aforementioned substances example in the painting of electrical panels and computers. Moreover, the use of lead-containing paints in Italy involves multiple limits and prohibitions regulated by national and regional regulations. For this reason, I deleted from the text the part regarding lead in paintings, and I updated the citation and the bibliography (page 15, line 158).

Reviewer 3 Report

title is not really attractive, I suggest something like "Bioaccumulation of trace elements in Stone Curlew (Burhinus oedicnemus, Linnaeus, 1758): a case study from Sicily (Italy)"

abstract is not informative

avoid keywords already in title as they are useless

introduction is quite generic in the beginning, but overall is sufficiently effective; the aim is not stated in the form of a testable hypothesis

Methods, a better and more detailed explanation about the statistical approach to sampling is necessary: I got the impression that it was based on preferential sampling

What was the post hoc test after Kruskal Wallis anova? perhaps the Dunn test

kg with lower K case througgout the paper and figs

explain details of boxplots in figs 2 and 3, I argue median and IQR as well as 1.5*IQR for whisker plus outliers: I would prefer seeing mean and SD/SE values since parametric anova was run

As is a metalloid throughout the paper

the discussion is quite long

conclusions are sufficiently supported by the data

the English language requires great improvement

overall the paper is nice, but what is puzzling me is the choice of the journal: I cannot see any molecular link, probably a more environment related journal would be much more appropriate (but I am confident this is an editorial choice)

Round 2

Reviewer 2 Report

Thanks for your replies

They all fullfilled my doubt.

So I consider the MS is worth of publication in the form it is.